# Isotope Fingerprints of Common and Tartary Buckwheat Grains and Milling Fractions: A Preliminary Study

**DOI:** 10.3390/foods11101414

**Published:** 2022-05-13

**Authors:** Lovro Sinkovič, Nives Ogrinc, Doris Potočnik, Vladimir Meglič

**Affiliations:** 1Crop Science Department, Agricultural Institute of Slovenia, Hacquetocva Ulica 17, SI-1000 Ljubljana, Slovenia; vladimir.meglic@kis.si; 2Department of Environmental Sciences, Jožef Stefan Institute, Jamova Cesta 39, SI-1000 Ljubljana, Slovenia; nives.ogrinc@ijs.si (N.O.); doris.potocnik@ijs.si (D.P.); 3Jožef Stefan International Postgraduate School, Jamova Cesta 39, SI-1000 Ljubljana, Slovenia

**Keywords:** *Fagopyrum*, hulls, semolina, isotopic signature, light flour, conventional, organic

## Abstract

The grains and milling fractions of common buckwheat (*Fagopyrum esculentum* Moench) and Tartary buckwheat (*Fagopyrum tataricum* (L.) Gaertn.) are widely used for both industrial and small-scale food and non-food products. This paper represents a preliminary study of the isotopic signature (*δ*^13^C, *δ*^15^N, and *δ*^34^S) to differentiate between buckwheat species (common vs. Tartary), organic and conventional cultivation farming, and different buckwheat fractions (light flour, semolina, and hulls) obtained by a traditional cereal stone-mill. Stable isotope ratios were analyzed using an elemental analyzer coupled to an isotope ratio mass spectrometer (EA/IRMS). The results indicated that *δ*^13^C, *δ*^15^N, and *δ*^34^S values could be used to verify the origin and production practices of buckwheat and even its products.

## 1. Introduction

Among the exploitable techniques, stable isotope-ratio signatures (*δ*^2^H, *δ*^13^C, *δ*^15^N, *δ*^18^O, and *δ*^34^S) are currently leading the way in food authenticity and traceability [1,2] in three main areas of application, namely (i) detection of adulteration; (ii) verification of geographical origin; and (iii) identification of mode of production, i.e., organic vs. conventional farming systems. Whereas conventional farming uses large quantities of synthetic fertilizer, organic farming relies heavily on organic fertilizers, such as animal manures and composts to maintain productivity. One of the significant differences between organic and synthetic fertilizers is their nitrogen isotope signatures (*δ*^15^N). It is expected that synthetic fertilizers have *δ*^15^N values close to atmospheric N_2_, while organic fertilizers are generally enriched in the heavier ^15^N isotope. The primary process leading to ^15^N enrichment of the substrate is NH_3_ volatilization. Therefore, it may be possible to differentiate organically and conventionally produced crops by looking at differences in their *δ*^15^N values [3,4]. However, there are at least two prerequisites for the successful application of a stable isotope differentiation technique: (i) isotopic fractionation caused by physical, chemical, or biochemical processes must be known, and (ii) there must exist a relevant database on which to perform a proper statistical evaluation [5]. Other analytical techniques and parameters have been studied to verify the provenance of regional foods and even type of production by gas and liquid chromatography and ‘fingerprinting’ or by chemical profiling based on ^1^H NMR (using hydrogen-1 as target atom), near Infra-Red and Fluorescence spectroscopy, and sensor and DNA technology [6,7]. These techniques can be powerful tools for determining food origin by analyzing specific characteristics of a raw material and product influenced by geographically or agri-production specific factors. Additionally, combining analytical techniques could be more beneficial than relying on one single method [6]. For example, NMR profiling has often been used with multi-isotopic and trace element analysis [7]. All of these methods have advantages and limitations, for example, isotope ratio mass spectrometry (IRMS) uses costly instrumentation, and the speed of the analysis is moderate. In addition, a common requirement is a need for an extensive quality assured database of authentic food samples, which is costly and time-consuming to construct [5].

Cereals have been the staple food of the major civilizations for over 8000 years and still account for a sizeable fraction of EU agricultural products. The current situation indicates that access to raw materials, including cereals, may become limited, and these materials may decline in quality. The quality assessment and authentication of cereals is of primary importance owing to the development of products with specific regional and varietal characteristics and the increasing demand for high-quality products in terms of their geographical origin [8]. Scientists have also used the stable isotope approach and elemental composition to classify wheat and other cereals into different categories and cultivation regions across Europe: north, south, the Atlantic Ocean, and the Mediterranean Sea [9]. For instance, Longobardi et al. [10] mainly used nuclear magnetic resonance (NMR) and isotope ratio mass spectrometry (IRMS) to characterize a wheat and wheat product’s geographical and varietal origin. These techniques provide a “fingerprint” of the foodstuffs, which can detect certain fraudulent practices and authenticate the geographical origin by comparison with authentic samples. In addition, these techniques can also provide an efficient means to enforce the restricted rules associated with Protected Designation of Origin (PDO)-labelled products [6,7].

Stable isotope ratio analyses (*δ*^13^C, *δ*^15^N, and *δ*^2^H, alone or with ^87^Sr/^86^Sr) have also been used to identify the geographical origin of winter wheat in China [11]. A further study by the same authors uses *δ*^2^H values of soil water during the three growth periods and rainwater, groundwater, and defatted wheat in the maturity stage to provide evidence for tracing its geographical origin [12]. Soil, water and crop elemental and isotopic composition has also been used to differentiate the provenance of Argentinean wheat [13]. A successful differentiation between three different regions (Buenos Aires, Córdoba, and Entre Ríos) was observed, where the most important isotope parameter was the *δ*^13^C value of wheat. *δ*^13^C values were also promising when the geographical origin of Indian wheat was studied, while the differences in the *δ*^15^N values from different states were insignificant [14]. Potential future authenticity issues are likely to come from new products becoming available. For example, pseudo-cereals such as buckwheat (*Fagopyrum*) are becoming increasingly popular due to perceived health benefits [15,16,17]. Since these products command a higher price, there is a distinct possibility that adulteration or mislabeling will occur.

Localized production of buckwheat grains is generally influenced by environmental factors and could, along with regional physical features such as geology, soil, and water, and agricultural practice (e.g., organic or conventional), resulting in characteristics isotope signatures. Consequently, the variability in isotopic compositions in buckwheat grains and their milling fractions creates a unique identifier that links to their cultivation origin [18]. Despite numerous publications concerning buckwheat cultivation and nutritional composition, none of them includes the isotopic fingerprints of buckwheat grains and their milling fractions to the best of our knowledge. Two species of the genus *Fagopyrum* are mainly cultivated and consumed worldwide: common buckwheat (*Fagopyrum esculentum* Moench) and Tartary buckwheat (*Fagopyrum tataricum* (L.) Gaertn.) [15]. Over the past ten years, the annual production of buckwheat has been between 1.5 and 3 million tons worldwide [19], while although Tartary buckwheat production is quantitatively less, its production probably most likely already accounts for a few percent of total production.

The objectives of the present study were (i) to analyze the isotopic signature of the whole grains, hulls, semolina, and light flour of common and Tartary buckwheat fractions obtained by a traditional stone-milling and harvested in three consecutive years; (ii) to compare the isotopic signature of the whole grains of two buckwheat species produced on organic and conventional fields; and (iii) to compare the isotopic signature of the whole grains of different buckwheat genetic resources (accessions/cultivars). Although buckwheat is recognized as nutritionally valuable, the isotopic composition of grains and milling fractions is entirely unknown. Therefore, the study aimed to evaluate and compare the stable isotope composition of individual fractions of the two buckwheat species. We are aware that the number of samples is limited; however, we believe that the information provided by the paper is essential for establishing an appropriate database of authentic buckwheat samples that, to our knowledge, does not currently exist.

## 2. Materials and Methods

### 2.1. Plant Material

Common buckwheat (*Fagopyrum esculentum* Moench; population from the Dolenjska region—CB_Eva) and Tartary buckwheat (*Fagopyrum tataricum* (L.) Gaertn.; traditional Slovene population—TB_Doris) were produced organically as a catch crop in the Posavje region (46°02′ N 15°13′ E, 194 m a.s.l., subcontinental climate) in a moderate soil and harvested in three consecutive years from 2012 to 2014. Seven common buckwheat (CB_Darja, CB_Darja Semenarna, CB_212, CB_Čebelica, CB_SUNOR 2007/41, CB_SUNOR 2010-14 and CB_Bamby) and ten Tartary buckwheat (TB_26, TB_96, TB_61, TB_66, TB_156, TB_115, TB_116, TB_65, TB_213 and TB_29) accessions were obtained from the Slovenian plant gene bank; produced conventionally as a main crop at the Infrastructure Center Jablje, Agricultural Institute of Slovenia, Slovenia (46.151°N 14.562°E, 304 m a.s.l., subalpine climate); and harvested in 2014. No fertilizers, herbicides, or fungicides were used before or during cultivation. After harvest, the grains were dried in a wooden drying chamber with ventilation at ambient temperature to reduce the moisture content to below 13%.

### 2.2. Milling Process and Sample Preparation

Over three consecutive years, common and Tartary buckwheat whole grains produced organically were further processed. A traditional stone mill with the capacity of 15 kg h^−1^ was used to obtain three milling fractions: hulls, semolina, and light four. Each fraction was manually checked after separation on mill sieves. The samples were stored at room temperature and low relative humidity in paper bags. The grains produced conventionally were analyzed as whole grains. Before analysis, the whole grains and milling fractions of all the samples were homogenized in a laboratory ball mill (Retsch MM400).

### 2.3. Stable Isotope Ratios

Stable isotope ratios of C, N, and S of powdered samples were determined simultaneously using an IsoPrime100-Vario PYRO Cube (OH/CNS) Pyrolyzer/Elemental Analyzer coupled with an isotope ratio mass spectrometer–IRMS (IsoPrime, Cheadle Hulme, UK). For analysis, 4 mg of sample and 4 mg of tungsten (VI)-oxide powder (Elementar Analysensysteme GmbH) were weighed directly into a tin capsule (Sercon, Crewe, UK). Tin capsules were closed with tweezers and placed into the automatic sampler of the elemental analyzer. Isotope data were expressed in *δ*-notation (‰) as follows: *δ*(‰) = [(R_sample_/R_standard_)–1] × 1000, where R is the ratio between the heavier and the lighter isotope (^13^C/^12^C, ^15^N/^14^N, ^34^S/^32^S) in the sample and standard, respectively. Values are reported relative to the following international standards: the Vienna Pee Dee Belemnite (VPDB) for carbon, atmospheric N_2_ (AIR) for nitrogen, and the Vienna Canyon Diablo Troilite (VCDT) for sulfur. The results for carbon were normalized against the international and laboratory reference materials: Wheat Flour Standard B2157 (Elemental Microanalysis Ltd., Okehampton, UK) with *δ*^13^C = −27.21 ± 0.13‰ and casein protein standard CRP-IAEA-2013 with *δ*^13^C = −20.30 ± 0.09‰. The reference material Protein (Casein) Standard B2155 (Elemental Microanalysis Ltd., Okehampton, UK) with *δ*^13^C = −26.98 ± 0.13‰ was used for quality control material and analyzed through the sequence, while the results for nitrogen and sulfur were normalized against the following international reference materials: Wheat Flour Standard B2157 and Protein (Casein) Standard B2155 with *δ*^15^N values of 2.21 ± 0.17‰ and 5.83 ± 0.08‰, respectively, and with *δ*^34^S values of −2.38 ± 0.80‰ and 6.18 ± 0.80‰, respectively, for sulfur. The laboratory reference material casein protein CRP-IAEA-2013 with *δ*^15^N = 5.62 ± 0.19‰ and *δ*^34^S = 4.18 ± 0.74‰ was used for quality control and regularly analyzed throughout the sequence. Each sample was measured at least twice. The reproducibility was ± 0.2‰ for *δ*^13^C and ± 0.3‰ for *δ*^15^N and *δ*^34^S.

## 3. Results and Discussion

Table 1 shows the isotopic signature of whole grains and milling fractions (hulls, semolina, and light flour) of common and Tartary buckwheat harvested in three consecutive years. Table 2 shows the isotopic signature of whole grains of common and Tartary buckwheat produced conventionally and organically. Statistical evaluation includes only univariate statistics using Kruskal–Wallis tests to compare isotopic parameters between different milling fractions, buckwheat variety, production year, and production regime. The results are collected in Table 3.

The *δ*^13^C values ranged from −31.1 to −27.7‰, while *δ*^15^N values ranged from 6.1 to 9.8‰. No statistically significant differences between different milling fractions, buckwheat variety, or the year of production (2012 and 2013) were observed in both parameters (Table 3). The *δ*^13^C values reported in our study are also lower and *δ*^15^N values higher than wheat samples reported in the literature. For example, Luo et al. [20] found *δ*^13^C values ranging from −25.7 to −22.3‰ and *δ*^15^N values from 1.9 to 7.7‰. Their study compared wheat samples from Australia, the USA, Canada, and China to discriminate the geographical origin of wheat. Rashmi et al. [14] recorded *δ*^13^C values between −27.8 and −24.6‰ and *δ*^15^N values between 1.3 and 3.8‰ for Indian wheat. They could differentiate the production region of Indian wheat using only their *δ*^13^C values. The study performed by Bontempo et al. [21] of durum wheat (*Triticum durum*) from four Italian regions (Basilicata, Molise, Emilia-Romagna, and Tuscany) over two years reported *δ*^13^C and *δ*^15^N values in ranging from −27.4 to −22.6‰ and from −0.4 to 10.6‰, respectively [21], with the high *δ*^15^N values are probably related to the higher application rate of animal manure. Our data exhibit lower *δ*^13^C values and are also lower than the German durum samples (−28.0 to −26.2‰) [22], indicating that besides climate, conditions such as the wheat variety, composition of soil and soil moisture can influence ^13^C distribution in the plant. Generally, *δ*^13^C values in plants reflect the signature of the soil organic C on the surface horizon of undisturbed (virgin) soils. In contrast, *δ*^15^N values are related to the available N sources rather than to soil organic N. In our case, the *δ*^15^N values of buckwheat indicate the possible presence of organic fertilizers. 

Alternatively, *δ*^34^S values ranging from 5.2 to 8.0‰ statistically differ between milling fractions (Table 3). Regarding *δ*^34^S, Italian wheat samples ranged from −25.2 to +8.9‰ [21], whereas German samples ranged from 3 to 6‰ [22]. Schmidt et al. [22] asserted that the *δ*^34^S value is affected significantly by the local geology and soil conditions. The lowest values found within the Italian samples were those from Tuscany, where high amounts of volcanic sulfur are present in the soil. It is interesting to note that *δ*^34^S values in our buckwheat samples are comparable with that of truffles (5.4 to 8.8‰) obtained from the same Dolenjska region [23]. 

Furthermore, the highest *δ*^34^S values (≤8.0‰) were obtained in hulls, indicating the heavier S isotopes’ preferential location. Plants take up S primarily as the sulfate anion, SO_4_^2−^ [24] and since there is little or no fractionation of the sulfur isotopes in plant metabolism, plants have *δ*^34^S values reflective of those in rainwater. However, sulfur has multiple biological roles, and our statistical differentiation in *δ*^34^S values between hulls and semolina can be explained by different sulfur assimilation. For example, in developing seeds of *M. truncatula*, the transcriptomic analysis revealed that, within the seed tissues, sulfur assimilation takes place by two distinct pathways. In one pathway, sulphate enters the embryo, is reduced and used for the biosynthesis of cysteine (Cys) that becomes incorporated into proteins. In another pathway, sulfate enters the seed coat and endosperm. The latter is preferentially involved in the biosynthesis of defense-related sulfur compounds or remains in sulfate form [25]. Pongrac et al. [26] also found that the outer layer of buckwheat is enriched in sulfur. Thus, the lower content of S enriched in lighter S isotopes is located in the semolina fraction.

The statistically relevant difference in the crop management system was observed for *δ*^15^N and *δ*^34^S (Table 3), with higher values obtained in organically produced buckwheat (Table 2) probably from using animal manure. Thus far, no study contains the stable isotope data on buckwheat, while the isotopic values for other wheat varieties produce contrasting data. For example, for spring barley (*Hordeum vulgare*), there were no significant differences in the *δ*^13^C values between samples from the two management systems, while *δ*^18^O and *δ*^15^N values varied significantly [27]. Specifically, the mean N and O ratios for the conventionally grown grains were 2.1 ± 0.3‰ and 14.7 ± 1.4‰, respectively, while for the organic samples, the mean N ratio was 3.4 ± 0.5‰ and the mean O ratio was 17.3 ± 0.6‰. Here, the *δ*^15^N values are lower than our buckwheat data. Chemometrics showed no clustering, and the authors suggested that further studies using larger sample sizes and more variables could improve the discrimination between the crop management systems. Gatzert et al. [28] found a significant variation in bulk *δ*^13^C values between the two management systems for common wheat (*Triticum aestivum*), while *δ*^15^N and *δ*^34^S showed no statistically significant difference.

Bontempo et al. [21] found no significant differences between the *δ*^13^C, *δ*^15^N, *δ*^18^O, and *δ*^34^S values of organic and conventionally produced Italian durum wheat. However, the discrimination between the two farming systems was improved if the evaluation focused on only one region, which is the case with our buckwheat samples.

Moreover, part of the Italian samples was milled into flour and processed into dry pasta by water addition. The results were surprising and revealed that none of the *δ*^2^H, *δ*^13^C, *δ*^18^O, *δ*^15^N, and *δ*^34^S values have been changed by the pasta production process. Thus, the isotope signature from the raw material (wheat) to the final product (pasta) has been retained despite water addition and pasta drying processes. This fact suggests that the pasta made from buckwheat can also retain the original isotopic signature from buckwheat.

## 4. Conclusions

This preliminary study includes the first multi-isotopic analysis of two buckwheat species and their fractions that can be used in authenticity studies. The *δ*^13^C and *δ*^15^N values show no statistically significant differences between the buckwheat type, milling fraction, or year of production, while *δ*^34^S values and sulfur content were higher in the hulls. This difference can be explained by different sulfur assimilation. The isotope parameters effectively discriminated the two farming systems (conventional vs. organic). However, this part needs to be further explored using larger sample sizes from different regions and other analytical techniques and parameters (e.g., multi-elemental or metabolomic fingerprinting).

## Figures and Tables

**Table 1 foods-11-01414-t001:** The isotopic signature of whole grains and milling fractions (hulls, semolina, and light flour) of common and Tartary buckwheat individual samples harvested during 2012–2014.

Year	Buckwheat Species	Fraction	*δ*^13^C (‰)	*δ*^15^N (‰)	*δ*^34^S (‰)
I	Common	Whole grains	−29.5	8.4	6.7
		Hulls	−29.8	7.8	6.6
		Semolina	−29.3	8.5	5.2
		Light flour	−29.1	8.5	5.4
	Tartary	Whole grains	−28.8	7.7	5.5
		Hulls	−29.4	6.7	7.7
		Semolina	−28.8	7.8	5.2
		Light flour	−28.2	7.5	6.7
II	Common	Whole grains	−30.2	6.4	5.4
		Hulls	−31.1	6.1	10.6 *
		Semolina	−30.0	6.2	5.0
		Light flour	−29.6	6.5	5.2
	Tartary	Whole grains	−28.4	8.9	8.0
		Hulls	−28.9	8.0	7.9
		Semolina	−28.6	9.0	5.5
		Light flour	−27.9	8.7	7.5
III	Common	Whole grains	−30.3	9.3	7.2
	Tartary	Whole grains	−27.7	9.8	5.5
Range			(−31.1)–(−27.7)	6.1–9.8	5.2–8.0

* Small peak (not considered in the further valuation).

**Table 2 foods-11-01414-t002:** The isotopic signature of whole grains of eight common and eleven Tartary buckwheat genetic resources produced conventionally and organically at Infrastructure Center Jablje in 2014.

Year	Buckwheat Species	Accession Name	Cultivation Field	*δ*^13^C (‰)	*δ*^15^N (‰)	*δ*^34^S (‰)
III	Common	CB_Eva	Organic	−30.3	9.3	7.8
		CB_Darja	Conventional	−28.6	5.0	4.6
		CB_Darja Semenarna		−30.0	5.3	4.7
		CB_212		−28.1	4.4	5.3
		CB_Čebelica		−29.6	6.0	3.6
		CB_SUNOR 2007/41		−29.3	5.2	5.1
		CB_SUNOR 2010-14		−29.3	4.6	4.6
		CB_Bamby		−29.3	4.0	4.8
	Range			(−30.3)–(−28.1)	4.0–9.3	3.6–7.8
	Tartary	TB_Doris	Organic	−27.7	9.8	5.8
		TB_26	Conventional	−29.3	3.5	3.9
		TB_96		−29.6	3.4	3.9
		TB_61		−29.4	3.7	3.7
		TB_66		−29.4	3.3	3.6
		TB_156		−29.6	3.7	4.0
		TB_115		−29.3	4.4	4.4
		TB_116		−30.0	4.4	4.3
		TB_65		−30.1	4.1	4.2
		TB_213		−29.5	3.8	3.3
		TB_29		−29.2	4.3	4.1
	Range			(−30.1)–(−27.7)	6.1–9.8	3.6–5.8

**Table 3 foods-11-01414-t003:** The univariate statistical evaluation results for selected parameters (*p*-values; confidence interval 95%). *p*-values in bold show a significant difference.

Parameter	Milling Fractions	Variety	Season	Type of Production
δ^13^C (‰)	*p* = 0.463	*p* = 0.550	*p* = 0.902	*p* = 0.422
δ^15^N (‰)	*p* = 0.402	*p* = 0.519	*p* = 0.156	***p* = 0.003**
δ^34^S (‰)	***p* = 0.035**	*p* = 0.382	*p* = 0.055	***p* = 0.005**

## Data Availability

The data presented in this study are available from the corresponding author upon request.

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
