# Peer review of "Isotope Fingerprints of Common and Tartary Buckwheat Grains and Milling Fractions: A Preliminary Study"

_foods, 2022, doi:10.3390/foods11101414_

Round 1
Reviewer 1 Report
The current short communication reports on the isotopic analysis data of common and Tartary buckwheat in an effort to authenticate the samples according to geographical origin, cultivation practises and processing. The communication provides the literature new data regarding common and Tartary buckwheat. It is well intoduced in the Foods journal and is of interest, in my opinion, for the readers. The English language is very good. There are minor corrections within the attached pdf. My recommendation is minor revision.

Reviewer 2 Report
The manuscript “Isotope Fingerprints of Common and Tartary Buckwheat Grains and Milling Fractions: Preliminary Study” provides a valuable approach regarding the isotopic signature (δ13C, δ15N, and δ34S) to differentiate between different buckwheat species and also, organic and conventional cultivation system.
Even if, at the beginning the paper is well organized and presents significance of content, the results and discussion section is poorly presented, being based mainly on comparison with other types of wheat presented in other papers.
Also, the authors affirmed that no statistically significant differences were observed between the different milling fraction but no statistical analysis is provided.
Even though in the first part of the article is mentioned the fact that a limited number of samples were analyzed, in the other sections no information is presented regarding the number of samples, for example if the values presented in table 1 are for individual samples or an average.
At the same time, the authors mention that an investigation of the influence of the harvest year has been carried out, but this is not discussed / interpreted in the results and discussions section, just mention in the scope and conclusions section.
Although the data presented are important for the authentication area of different food products based on isotopic fingerprinting, the construction of authentic databases requires much more real samples in order to exclude different variations given by different environmental or genetic factors.
Reviewer 3 Report
This paper provided a preliminary study on the isotope fingerprints of common and Tartary buckwheat grains and milling fractions. This study was novel and significant. However, there are still some things done to improve this manuscript. Below are some comments that may further improve the quality of this manuscript. And this manuscript will be recommended after major revisions.
- In the “Introduction”, the advantages and disadvantages of the stable isotope ratio analyses should be further discussed and compared with some other identification methods.
- In table 1, the isotopic signatures of whole grains and milling fractions (hulls, semolina, light flour) of common and Tartary buckwheat harvested in the first two years were provided. However, why was the isotopic signature of only whole grains given in the third year?
- In line 27, the “animal manures, composts” should be revised as “animal manures and composts”.
- In line 66, the “result in” should be revised as “resulting in”.
- In line 134 and 155, the “and” should be added after “hulls, semolina,”.
- In line 191, the “,” should be added after “δ15N”.
Round 2
Reviewer 2 Report
The submitted manuscript gives valuable results and it might be helpful for the researchers in the field of buckwheat authenticity based on stable isotopes. Also, the manuscript has been improved according to my observations.
In my view, the manuscript can be published.